Nests of red wood ants (Formica rufa-group) are positively associated with tectonic faults: a double-blind test

http://orcid.org/0000-0002-3901-8713 Del Toro Israel 1 israel.deltoro@lawrence.edu
Berberich Gabriele M. 2
http://orcid.org/0000-0002-9566-3813 Ribbons Relena R. 1
Berberich Martin B. 3
Sanders Nathan J. 4
http://orcid.org/0000-0003-4151-6081 Ellison Aaron M. 5
1 Department of Biology, Lawrence University , Appleton, WI , USA
2 Faculty of Electrical Engineering and Information Technology, Technical University of Dortmund , Dortmund , Germany
3 IT-Consulting Berberich , Erftstadt , Germany
4 Rubenstein School of Environment and Natural Resources, University of Vermont , Burlington, VT , USA
5 Harvard Forest, Harvard University , Petersham, MA , USA
Gillespie Joseph
Electronic publication date: 2017 Oct 12
Publication date: 2017
Volume: 5
Electronic Location ID: e3903
Received 2017 Jul 6; Accepted 2017 Sep 18
Copyright: © 2017 Del Toro et al.
Copyright year: 2017
Copyright holder: Del Toro et al.
License: This is an open access article distributed under the terms of the Creative Commons Attribution License, which permits unrestricted use, distribution, reproduction and adaptation in any medium and for any purpose provided that it is properly attributed. For attribution, the original author(s), title, publication source (PeerJ) and either DOI or URL of the article must be cited.
License URL: https://creativecommons.org/licenses/by/4.0/

Keywords: Double-blind, Tectonic faults, Formicidae, Species distributions, Clustering

Funding: National Science Foundation Postdoctoral Research Fellowship NSF grant DEB-1136646 National Science Foundation Dimensions of Biodiversity grant NSF-1136703 Israel Del Toro was supported by a National Science Foundation Postdoctoral Research Fellowship; Aaron M. Ellison was supported by NSF grant (DEB-1136646); Nathan J. Sanders was supported by a National Science Foundation Dimensions of Biodiversity grant (NSF-1136703). The funders had no role in study design, data collection, and analysis, decision to publish, or preparation of the manuscript.

==============================
Ecological studies often are subjected to unintentional biases, suggesting that improved research designs for hypothesis testing should be used. Double-blind ecological studies are rare but necessary to minimize sampling biases and omission errors, and improve the reliability of research. We used a double-blind design to evaluate associations between nests of red wood ants (Formica rufa, RWA) and the distribution of tectonic faults. We randomly sampled two regions in western Denmark to map the spatial distribution of RWA nests. We then calculated nest proximity to the nearest active tectonic faults. Red wood ant nests were eight times more likely to be found within 60 m of known tectonic faults than were random points in the same region but without nests. This pattern paralleled the directionality of the fault system, with NNE–SSW faults having the strongest associations with RWA nests. The nest locations were collected without knowledge of the spatial distribution of active faults thus we are confident that the results are neither biased nor artefactual. This example highlights the benefits of double-blind designs in reducing sampling biases, testing controversial hypotheses, and increasing the reliability of the conclusions of research.

Introduction

A central question for ecology—the study of the distribution and abundance of organisms—is why do organisms occur where they do? Explanations include relationships between organisms and specific environments, interspecific interactions, or random chance. All of these explanations have been suggested to apply to ants, one of the most widespread and abundant taxon on Earth (Hölldobler & Wilson, 1990; Lach, Parr & Abbott, 2010). Berberich & Schreiber (2013) and Berberich et al. (2016b) reported a seemingly peculiar positive spatial association between the geographically widespread, conspicuous red wood ants (Formica rufa-group, RWA) and seismically active, degassing tectonic faults. This work has been difficult to publish because reviewers have suggested that the authors are ignoring alternative explanations or are ignorant of the basic biology of ants. Such critiques are familiar to anyone who has proposed a new or controversial hypothesis, but it is indeed peculiar that ants would be associated with degassing tectonic faults.

Here, we confront the observations of Berberich and colleagues using a double-blind study. Double-blind studies, in which treatment assignments (or data collected) are concealed to researchers and subjects, are the most robust ones for testing any hypothesis, especially controversial ones, and increase the reliability of results and conclusions (Holman et al., 2015). Double-blind designs are routine in medical sciences, but rare in ecology (Kardish et al., 2015). To test more robustly the hypothesis that RWA nests are associated with active faults, we used a double-blind design in which myrmecologists who were unaware of this hypothesis or any published work on links between RWA and seismic activity (IDT and RRR) were sent into the field to map RWA nests. Simultaneously, maps of active tectonic faults in the region were obtained and organized by geoscientists (GMB and MBB) without any knowledge of the field data. With these two independently collected datasets, we then asked whether ants were positively associated with tectonic faults.

Materials and Methods

Sampling design and data collection

With no prior knowledge, IDT and RRR surveyed two regions of the Jutland Peninsula of Denmark: Thisted in the north and Klosterhede in the south (Figs. 1A–1C). Both study areas are located within the Permian–Cenozoic Danish Basin, which was formed by crustal extension, subsidence, and local faulting (Nielsen, 2003). This basin is bounded in the north by the seismically active, NW–SE striking fault system of the Sorgenfrei-Tornquist-Zone (STZ in Fig. 1A) and in the south by the basement blocks of the Ringkøbing–Fyn High and the Brande Graben (RFH and BG in Fig. 1A). The dominating compressional stress field is orientated primarily NW–SE direction (Fig. 1D) but scatters in different regions (Gregersen, 2002; Helmholtz Centre Potsdam—GFZ German Research Centre for Geosciences, 2008).

Figure 1 (A) Map of Jutland Peninsula, highlighting the two sampling regions and major tectonic units.

Thisted (B) and Klosterhede regions (C) are shown in detail. Thisted region (north) and the Klosterhede region (south) are shaded in red. Blue points indicate absence of RWA from grid survey; red points indicate the location of RWA nests. (D) Distribution of faults in the Jutland Peninsula (after Vejbæk, 1997; Petersen et al., 2008; The Geological Survey of Denmark and Greenland, 2015) with red and blue points as in C and D.

The Thisted region (∼670 km2) included parts of the Thy National Park. The Klosterhede region (∼700 km2), included the Klosterhede plantation, the third largest forested area in Denmark. Landscapes and vegetation communities varied between the two sampling regions. Coastal dunes dominated the Thisted region, whereas a mix of grasslands, pine and oak forests, and conifer plantations dominated the Klosterhede region. Agricultural lands in both regions were primarily rapeseed plantations.

Before surveying for RWA nests, and with no prior knowledge of the spatial distribution of tectonic faults, the two regions were subdivided into ∼1,000 × 1,500 m grid cells. One hundred of the cells in each region were selected at random for mapping RWA nests, using the rnorm function in R (version 3.31). At each site, we used an adaptive sampling design to search for RWA nests. If no RWA nests were encountered within an initial 30 min sampling period, we considered RWA to be absent from the grid cell. However, if a RWA nest was encountered within an initial 30 min of searching, the survey was continued for an additional 30 min; this process was repeated until no new nests were found within the survey grid cell boundaries. The location of each RWA nest found was recorded using a Garmin Oregon 600 GPS unit (Garmin, Olathe, KS, USA); three individual worker ants were collected from every nest for subsequent species identification. Voucher specimens were deposited in the Natural History Museum of Denmark, Copenhagen.

GMB and MBB synthesized published data on geotectonic structures of the two study areas (data in Supplementary Online Material) with tectonic maps provided by Stig Pedersen (Geological Survey of Denmark) and the GEUS Map Server (The Geological Survey of Denmark and Greenland, 2015); they did so with no knowledge of the distributions of the RWA nest data collected by IDT and RRR.

Spatial data and analyses

Spatial clustering of RWA nests was examined with Ripley’s K (Ripley, 1977). The distance from each nest to the nearest fault line was calculated using the “distmap” function in the “spstat” library (Baddeley, Rubak & Turner, 2015) using R (version 3.3.1) (R Development Core Team, 2014). We then estimated ρ: the effect of the spatial covariate (i.e., distance to faults) on the spatial intensity of the locations of the ant nests and the locations of cells without ants (Baddeley et al., 2012). Finally, we used a Komlogrov–Smirnov (K–S) test to test if observed RWA nests were closer to faults than locations sampled (i.e., the center of the sampled grid cell) where no RWA nests were detected. We attempted to reduce sampling bias resulting from spatial autocorrelation by using a random sampling grid. The remaining spatial autocorrelation or clustering most likely is related to the polydomous and polygynous colony structure of many F. rufa-group species (Seifert, 2007).

Results and Discussion

RWA nests occur closer to fault lines than expected by chance

Red wood ant nests occurred in 28 of the 200 random grid cells (12 in the Thisted region and 16 in the Klosterhede region). When RWA occurred in a sampled grid cell, there were generally >1 nest; in total we detected 273 nests of Formica species. All but four (one Formica serviformica and three Formica fusca) were nests of F. rufa-group ants: 86% were nests of Formica polyctena and 12% were F. rufa. In both regions, RWA nests were spatially clustered according to Ripley’s K, but cells without ants were not spatially clustered (Fig. 1).

Covariance of RWA nests and faults was highest within 60 m of faults (Fig. 2A), and approached zero at greater distances. In contrast, there was no observable covariance between cells without RWA nests and their distance from faults (Fig. 2B). RWA nests were approximately eight times more likely to be found at distances <60 m from a fault than were cells without ants (K–S test, D = 0.373, P < 0.001).

Figure 2 (A) Correlation of ant distributions with tectonic fault zones in the region.

(B) Correlation of ant absences with tectonic fault zones in the region. (C–H) Correlations of ant distributions with direction patterns of tectonic fault zones in the pooled dataset.

The directionality of a fault also affected the covariance between the spatial intensity of RWA nests and their distance to faults (Figs. 2C–2H). Specifically, at distances <100 m from an active fault and relative to grid cells lacking ants, RWA nests were 10 times more likely along faults trending NNE–SSW (Fig. 2F) and up to eight times more likely on faults trending NW–SE or NNW–SSE (Figs. 2C and 2D). These directions are associated with the present-day main tectonic stress field and its scattering directions (Helmholtz Centre Potsdam—GFZ German Research Centre for Geosciences, 2008). In contrast, RWA were only two to four times more likely to faults trending NE–SW or WNW–ESE (Figs. 2E and 2G), and did not occur adjacent to faults trending ENE–WSW (Fig. 2H).

To address further whether polydomous colony structure of F. polyctena could have skewed our results, we repeated the analysis for F. rufa alone. Similar to the overall pattern, F. rufa tended to be clustered within 30 m from existing degassing faults (Fig. 3A). We also tested this pattern without considering individual nests but only the plots where RWA was present; once again the pattern remained consistent and showed a peak in association with faults at distance between 0 and 30 m (Fig. 3B). This suite of evidence leads us to propose a hypothesis that drives the observed biological pattern and should be tested in subsequent studies. Previous work has associated microhabitat availability to the spatial structure of ant colonies (Scharf, Fischer-Blass & Foitzik, 2011). In this study system, we argue that the underlying fault lines may be providing a warmer thermal microhabitat, which drives the spatial clustering of RWA colonies.

Figure 3 (A) Spatial correlation of Formica rufa with tectonic faults.

(B) Spatial correlation of grid units with RWA with tectonic faults.

On the use of double-blind studies in ecology

The scientific method emphasizes accurate, unbiased, and objective experiments or observations. Because research results can be biased by design or our underlying belief in the correctness of our hypothesis (confirmation bias: (Nickerson, 1998)), repeatable results and reliable conclusions require that investigators do as much as possible to minimize bias in all aspects of a research project (Rosenthal & Rosnow, 2007). Double-blind designs provide the gold-standard for unbiased experiments (Holman et al., 2015).

In the interest of avoiding bias and increasing the repeatability and reliability of ecological research, we suggest that the benefits of double-blind studies far outweigh the additional costs and logistical complications of creating blinded research teams. The need for multiple research teams leads directly to increased costs and additional project coordination. Trade-offs among personnel, sampling effort, and sampling intensity depend on available resources. In our study, for example, we reduced sampling effort by randomly, not exhaustively, sampling the ∼1,400 km2 of the pre-defined study regions. A second cost of a double-blind study such as ones focused on species occurrences is the general tendency to focus on where a species occurs, as opposed to where it does not. For example, most species distribution models are based only on “presence-only” data, as absences are rarely recorded (Berberich et al., 2016a). Yet as we have shown here, the samples of locations lacking RWA nests were crucial for determining whether RWA nests and fault systems had meaningful patterns of covariance.

Double-blind experiments remain rare in ecology (Holman et al., 2015) but their importance cannot be overestimated. Results and conclusions of double-blind studies are unlikely to be biased by the views and perspectives of the researchers themselves. Investment in appropriately replicated double-blind studies also may be more cost-effective because they rarely need to be repeated, even if the results are unexpected. Just as double-blind studies in medicine have led to reliable treatments that for injury and disease, double-blind studies in ecology will provide us with high-quality unbiased data of how the natural world is structured and is changing.

Supplemental Information

Supplemental Information 1 Supplementary R code.

Knitted R markdown file showing all analyses.

Click here for additional data file.

Supplemental Information 2 Formica distribution data.

Click here for additional data file.

Supplemental Information 3 Fault line dataset.

Click here for additional data file.

Supplemental Information 4 R Code for analyses.

Click here for additional data file.

Additional Information and Declarations

Competing Interests

Author Contributions

Data Availability

Aaron Ellison is an Academic Editor for PeerJ. Martin B. Berberich is an employee of IT-Consulting Berberich.

Israel Del Toro conceived and designed the experiments, performed the experiments, analyzed the data, contributed reagents/materials/analysis tools, wrote the paper, prepared figures and/or tables, reviewed drafts of the paper.

Gabriele Berberich conceived and designed the experiments, performed the experiments, contributed reagents/materials/analysis tools, wrote the paper, reviewed drafts of the paper.

Relena R. Ribbons performed the experiments, contributed reagents/materials/analysis tools, wrote the paper, reviewed drafts of the paper.

Martin B. Berberich performed the experiments, contributed reagents/materials/analysis tools, wrote the paper, reviewed drafts of the paper.

Nathan J. Sanders conceived and designed the experiments, contributed reagents/materials/analysis tools, wrote the paper, reviewed drafts of the paper.

Aaron M. Ellison conceived and designed the experiments, analyzed the data, contributed reagents/materials/analysis tools, wrote the paper, reviewed drafts of the paper.

The following information was supplied regarding data availability:

The R markdown file is knitted and presented as Supplemental Dataset Files.

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
