# Peer review of "Nests of red wood ants (Formica rufa-group) are positively associated with tectonic faults: a double-blind test"

_PeerJ, doi:10.7717/peerj.3903_

## Round 0.1 · original submission · Minor Revisions

I find this an interesting paper with a well-crafted experimental design. It is an important addition to the work already accomplished by some of the authors, and has potential to attract workers outside of the Hymenoptera community. Well done! Please carefully consider the concerns raised by BOTH reviewers, and in particular, how you may better explain some of the experimental approaches. The idea that nests might not be independent for F. polyctena should also be addressed. The relevance of the double-blind approach should also be more clearly delivered. Good luck with your revision!

·

Basic reporting

The authors have conducted a double-blind test to test the hypothesis that red wood ants occur in proximity to active tectonic faults. Former publications of some of the co-authors have shown this already. I have to admit – I was not aware that this hypothesis or the published results were criticized by the myrmecologist community. In this case I welcome this study with its rigorous experimental approach if it helps to corroborate former results in an unbiased way.

Experimental design

A potential weakness of the study is the polydomous colony structure of one of the ant species studied here (see comment 1). I have two points that I think the authors should clarify in this otherwise well-performed and well-presented study.
1. The authors state that the RWA nests were spatially clustered. Especially for F. polyctena this is no surprise as this ant is known to be highly polydomous (and polygynous) (e.g. in Seifert 2007 or the edited book on RWA ecology and conservation by Stockan and Robinson). Thus, the clustering of nests can to my mind be considered a pseudoreplication as nests are (at least in part) not independent of each other. What are the results for F. rufa alone in comparison to F. polyctena? Here this problem should not be as pronounced. Are the results comparable? Could you potentially do the analysis with the grid cells themselves (RWA present or absent)?
2. One point that is to my mind not made clear enough in the methods section is how random points in the same region were generated for comparison. Please add one or two sentences.

Validity of the findings

no comments

Reviewer 2 ·

Basic reporting

no comment

Experimental design

Line 20-22: How conducted study in this manuscript does help us to apply (design) double-blind ecological studies in “distribution” and “abundance” studies. As far as I understand the presented study is not related to any of these groups in their classical definition, at least. An example in the discussion/conclusion can help to understand the point authors try to address.

Validity of the findings

Line 94-100: How do authors deal with spatial autocorrelation in this analyze.

Line 125-149: I miss any discussion/conclusion on why ant nests are significantly closer to faults?

Additional comments

The manuscript has overall been written well. It seems authors have not been able to decide if the main topic is testing double-blind approach in the Ecology or an ecological experiment on relationship of ant nests and tectonic faults. Authors can the re-organize the structure in a new way that readers can follow the paper better.

---

## Round 0.2 · accepted · Accept

Dear Dr. Del Toro and colleagues:

I am delighted to report that your work has been deemed suitable for publication in PeerJ. Congratulations! Thank you for addressing all of the concerns of the reviewers, as the manuscript is much improved. This is a very interesting report, and we are excited to be publishing your work in PeerJ. I personally look forward to seeing it in print!

Best,

-joe